Assessment of intra- and inter-genetic diversity in tetraploid and hexaploid wheat genotypes based on omega, gamma and alpha-gliadin profiles

Al-Khayri Jameel M. jkhayri@kfu.edu.sa 1
Alwutayd Khairiah M. 2
Safhi Fatmah A. 2
Alqahtani Mesfer M. 3
Alshegaihi Rana M. 4
Abd El-Moneim Diaa dabdelmoniem@aru.edu.eg 5
Jain Shri Mohan 6
Eldomiaty Ahmed S. 7
Alshamrani Rahma 8
Abuzaid Amani Omar 8
Hassanin Abdallah A. 7
1 Department of Agricultural Biotechnology, College of Agriculture and Food Sciences, King Faisal University , Al-Ahsa , Saudi Arabia
2 Department of Biology, College of Science, Princess Nourah bint Abdulrahman University , Riyadh , Saudi Arabia
3 Department of Biological Sciences, Faculty of Science and Humanities, Shaqra University , Ad-Dawadimi , Saudi Arabia
4 Department of Biology, College of Science, University of Jeddah , Jeddah , Saudi Arabia
5 Department of Plant Production, (Genetic Branch), Faculty of Environmental and Agricultural Sciences, Arish University , El-Arish , Egypt
6 Department of Agricultural Sciences, University of Helsinki , Helsinki , Finland
7 Genetics Department, Faculty of Agriculture, Zagazig University , Zagazig , Egypt
8 Biology Department, Faculty of Science, King Abdulaziz University , Jeddah , Saudi Arabia
Kutlu Imren
Electronic publication date: 2023 Nov 7
Publication date: 2023
Volume: 11
Electronic Location ID: e16330
Received 2023 Mar 21; Accepted 2023 Sep 30
Copyright: ©2023 Al-Khayri et al.
Copyright year: 2023
Copyright holder: Al-Khayri et al.
License: This is an open access article distributed under the terms of the Creative Commons Attribution License, which permits unrestricted use, distribution, reproduction and adaptation in any medium and for any purpose provided that it is properly attributed. For attribution, the original author(s), title, publication source (PeerJ) and either DOI or URL of the article must be cited.
License URL: https://creativecommons.org/licenses/by/4.0/

Keywords: Genetic diversity, Durum wheat, Bread wheat, Gliadin, Bioinformatics, SDS-PAGE

Funding: Deanship of Scientific Research, Vice Presidency for Graduate Studies and Scientific Research, King Faisal University, Saudi Arabia GRANT4365 Princess Nourah bint Abdulrahman University Researchers Supporting Project number: PNURSP2023R318, Princess Nourah bint Abdulrahman University, Riyadh, Saudi Arabia This work was supported by the Deanship of Scientific Research, Vice Presidency for Graduate Studies and Scientific Research, King Faisal University, Saudi Arabia (Project No. GRANT4365), and the Princess Nourah bint Abdulrahman University Researchers Supporting Project number (PNURSP2023R318), Princess Nourah bint Abdulrahman University, Riyadh, Saudi Arabia. The funders had no role in study design, data collection and analysis, decision to publish, or preparation of the manuscript.

==============================
Durum and bread wheat are well adapted to the Mediterranean Basin. Twenty-three genotypes of each species were grown to evaluate the intra- and inter-genetic diversity based on omega (ω), gamma (γ) and alpha (α)-gliadin profiles. To achieve this purpose, the endosperm storage proteins (both gliadins and glutenins) were extracted from wheat grains and electrophoresed on sodium dodecyl sulfate (SDS)–polyacrylamide gels. The results of SDS-Polyacrylamide Gel Electrophoresis (SDS-PAGE) revealed nine polymorphic loci out of 16 loci with durum wheat genotypes and nine polymorphic loci out of 18 loci with bead wheat genotypes. The polymorphisms revealed by the SDS-PAGE were 56% and 50% in durum and bread wheat genotypes, respectively. Using the cluster analysis, the durum wheat genotypes were clustered into five groups, while the bread wheat genotypes were grouped into six clusters using un-weighed pair group mean analyses based on ω, γ, and α-gliadins profiles. The 46 durum and bread wheat genotypes were grouped into seven clusters based on the combined ω, γ, and α-gliadins profiles revealed by the SDS-PAGE. The in silico analysis determined the intra-genetic diversity between bread and durum wheat based on the sequences of ω, γ, and α-gliadins. The alignment of ω-gliadin revealed the highest polymorphism (52.1%) between bread and durum wheat, meanwhile, the alignment of γ and α-gliadins revealed very low polymorphism 6.6% and 15.4%, respectively. According to computational studies, all gliadins contain a lot of glutamine and proline residues. The analysis revealed that the bread wheat possessed ω and γ -gliadins with a lower content of proline and a higher content of glutamine than durum wheat. In contrast, durum wheat possessed α-gliadin with a lower content of proline and a higher content of glutamine than bread wheat. In conclusion, the SDS-PAGE, in silico and computational analyses are effective tools to determine the intra- and inter-genetic diversity in tetraploid and hexaploid wheat genotypes based on ω, γ, and α-gliadins profiles.

Introduction

Durum (Triticum turgidum L. spp. Durum Desf.) and bread (Triticum aestivum L.) wheat are the two main cereal species of the Mediterranean Basin. Bread wheat is mainly used to produce bread wheat flour, whereas durum wheat is used to produce semolina for pasta production. Wheat is the most essential cereal crop in the world in terms of production consumption for animal feed and human food. Almost 5% of wheat is also used for seed planting and other purposes (alcohol, glues, etc). Bread wheat has a hexaploid genome arrangement (2 n = 6x = 42) (AABBDD) while durum wheat has a  tetraploid genome arrangement (2 n = 4x = 28) (AABB).

One of the most important sources of proteins for human diet is gluten, a variety of foods have been developed to benefit from the qualities of wheat flour, such as its baking characteristics, dough rheology, and mixing properties. It is found that the quantities of gluten polymers in wheat alter its viscoelastic qualities, and the allelic variation in the composition of gliadins and HMW-glutenins is associated with variation in the bread quality (Payne, 1987; Shewry & Halford, 2002). However, few studies were conducted to evaluate the genetic diversity between and within durum and bread wheat genotypes using biochemical markers.

The recombination of genetic material during the inheritance process results in genetic diversity. studying of genetic diversity enables the determination of genetic traits correlated to important breeding objectives (Pour-Aboughadareh et al., 2022). Also, the assessment of genetic diversity is crucial in breeding experiments for selecting cultivars with higher diversity and better performance under a certain condition (Pour-Aboughadareh et al., 2018; Essa et al., 2023). The plant breeding can maximize the benefits of heterosis by using the genetic polymorphism information to select the parents to cross for cultivar development or in a hybrid combination (Nybom, Weising & Rotter, 2014; Sreewongchai, Sripichitt & Matthayatthaworn, 2021). Consequently, understanding of nature and the degree of genetic diversity in the resources at their disposal are usually needed for geneticists and breeders. 

The SDS-PAGE of seed protein is an approach to assess the genetic diversity and classify plant genotypes (Javaid, Ghafoor & Anwar, 2004; Iqbal, Ghafoor & Ayub, 2005; Khan et al., 2020). The assessment of the genetic variation between and within wheat genotypes provides a chance for plant breeders to develop new genotypes with desirable properties. The SDS-PAGE of seed protein is an approach to assess the genetic diversity and classify plant genotypes (Jan et al., 2019). The monomeric gliadins have a molecular weight of approximately 30–60 kDa and are soluble in aqueous alcohol solutions. Although some gliadin types may be associated with glutenin subunits, the γ-, α-, and β-gliadins all include disulfide bonds (Girard, 2022). The ω-gliadin is also named sulfur-poor prolamin (Rustgi et al., 2019) due to the absence of disulfide linkages. The amount of gliadins and glutenins (which make up 35 to 45% dw each) is affected during growth by both environment and genotype. According to Metakovsky et al. (2021) the gliadins and glutenins are heritated at several loci on each genome A, B, and D. The SDS-PAGE approach is mainly utilized for the electrophoresis of seed proteins on polyacrylamide gel (Khan et al., 2010). The banding patterns of seed proteins have been utilized to investigate the evolutionary relationships between various crop species (Hamouda, 2019). Priority for improving varieties should be given to gathering these genetic resources and assessing genetic variation inter- and intra-genotypes (Sadia et al., 2009). There are further benefits to using the SDS-PAGE in actual plant breeding, including its simplicity and low cost (Sadia et al., 2009). Also, the SDS-PAGE was used to assess the degree of polymorphism for durum and bread wheat based on variation in the loci coding for the high and the low molecular weight glutenin subunits (Ruiz & Giraldo, 2021; Visioli et al., 2021; Cao et al., 2022). The allelic variation at glutenin loci is lower than for gliadin loci (Dai et al., 2022). The variety of gliadins has an impact on the extensibility, tenacity and strength of dough (Rodríguez-Quijano et al., 2019). The gliadins are crucial in giving the dough extensibility. It has been determined that some gliadin alleles are positively correlated with dough strength and extensibility (Islam et al., 2019).

Cluster analysis is utilized to group genotypes depending on the features they exhibit. So, genotypes with similar characteristics are mathematically grouped into the same cluster (Hair et al., 1995). As a result, the genotype clusters that emerge should have high levels of internal homogeneity (inside the cluster) and high levels of external heterogeneity (between clusters). Therefore, when represented geometrically, genotypes within a cluster will be closer together while those of other clusters will be farther away (Hair et al., 1995). In general, clustering techniques fall into two categories: (i) distance-based methods and (ii) model-based methods (Johnson & Wichern, 2002). Distance-based clustering methods include two groups: hierarchical and nonhierarchical. The hierarchical clustering method is more globally used to determine the genetic variation in plant varieties. The unweighted paired group method using arithmetic averages (UPGMA) is the most widely used clustering algorithm among many agglomerative hierarchical approaches (Sneath & Sokal, 1973; Panchen, 1992; Al-Khayri et al., 2022; Al-Khayri et al., 2023a). The main objective of this work is to evaluate the genetic diversity between and within durum and bread wheat genotypes to divide those genotypes into groups depending on the ω, γ, and α-gliadins profiles in addition to multiple alignment and computational analyses of ω, γ, and α-gliadins to determine the endosperm protein characteristics and the type of the amino acids. It is also of great value to evaluate the possible use of gliadin banding profiles to predict quality in plant breeding.

Materials and Methods

Plant materials

This study included 23 durum wheat and 23 bread wheat advanced inbred lines obtained from CIMMYT and ICARDA (Table 1) to assess the genetic diversity between and within the bread and durum wheat genotypes based on ω, γ, and α-gliadin banding profiles. The evaluated advanced inbred lines were chosen based on their superiority in dough quality from previous preliminary screening trial (unpublished data). The seeds were planted at the farm of the Faculty of Agriculture, Zagazig University (Longitude: 31°30′07″E, Latitude: 30°35′15″N, height above sea level: 16 m (52 ft)), in the growing season 2021/2022. The seeds were sown during the third week of November. The fertilization, irrigation, pest control and weed control were applied.

Table 1 List of durum and bread wheat genotypes with their pedigree.

Genotype (G)	Pedigree	Origin	
	Bread wheat		
G1	Qamar-4Cmss97m03159t-040y-0b-0ap-2ap-…	CIMMYT	
G2	D67.2/Parana66.270//Ae.Squarrosa (320)/3/…(synthetic)	CIMMYT	
G3	Cno79//Pf70354/Mus/3/Pastor/4/Bav92/5/Mılan	CIMMYT	
G4	Babax/Ks93u76//Babax/3/2*Sokoll Cmsa06m	CIMMYT	
G5	D67.2/Parana 66.270//Ae.Squarrosa (320)/3/(synthetic)	CIMMYT	
G6	Krıchauff/2*Pastor/4/Mılan/Kauz//Prınıa/3/Bav	CIMMYT	
G7	Heılo//Sunco/2*Pastor Cmsa06y00492s-040zty-	CIMMYT	
G8	Chıh95.7.4//Inqalab 91*2/Kukuna Ptss06ghb..	CIMMYT	
G9	Kachu #1/Kırıtatı//Kachu Cmss06y00778t-099..	CIMMYT	
G10	Saual/Yanac//Saual Cmss06y00783t-099topm..	CIMMYT	
G11	Prl/2*Pastor*2//Fh6-1-7 Cmss06y00793t-099…	CIMMYT	
G12	Frncln/Rolf07cmss06b00013s-0y-099ztm-099y	CIMMYT	
G13	Becard/Kachu Cmss06b00169s-0y-099ztm-099	CIMMYT	
G14	Becard/AkurıCmss06b00411s-0y-099ztm-099y	CIMMYT	
G15	Rolf07*2/5/Reh/Hare//2*Bcn/3/Croc_1/Ae….	CIMMYT	
G16	Usher-16 Crow’s’/Bow’s’-1994/95//Asfoor-5…	CIMMYT	
G17	Croc_1/Ae.Squarrosa(213)//Pgo/3/Cmh81.38/2 (synthetic)	CIMMYT	
G18	Chen/Aegılops Squarrosa (Taus)//Bcn/3/Bav92. (synthetic)	CIMMYT	
G19	Mısket-12-Btı735/Achtar//Asfoor-1ıcw01-…	CIMMYT	
G20	Rebwah-12/Zemamra-8-Rebwah-12/Zemamra	CIMMYT	
G21	HAMAM-4/ANGI-2//PASTOR-2	ICARDA	
G22	SEKSAKA-7/3/SHUHA-2//NS732/HER	ICARDA	
G23	QAFZAH-2/FERROUG-2//ZEMAMRA-8	ICARDA	
	Durum wheat		
G1	M84859	ICARDA	
G2	M141979	ICARDA	
G3	M141982	ICARDA	
G4	M141994	ICARDA	
G5	M141995	ICARDA	
G6	M142005	ICARDA	
G7	M142017	ICARDA	
G8	M142025	ICARDA	
G9	M142038	ICARDA	
G10	M142045	ICARDA	
G11	M142069	ICARDA	
G12	M142070	ICARDA	
G13	E90040/MFOWL13	ICARDA	
G14	SRN1/LARU/3/YAV /FGO//ROH/4/LICAN	ICARDA	
G15	TANTLO//CREX/ALLA/3TANTLO	ICARDA	
G16	Lgt3/4/Bcr/3/Chi//Gta/Stk	ICARDA	
G17	Bcr//Memo/goo	ICARDA	
G18	Bcr//Memo/goo/3/Stjy	ICARDA	
G19	D68-1-93A-1A//Ruff/Fg/3/Mtl-5/4/Lahn	ICARDA	
G20	Bcr//fg/snbipe/3/Gdovz 578/swan//Ddra2	ICARDA	
G21	Villemur/3/Lahn//gs/stk/4/Dra2/Bcr	ICARDA	
G22	Terbo 197-4	ICARDA	
G23	Stj3//Bcr/LKS4	ICARDA	

Protein extraction and electrophoresis

Extraction of endosperm proteins and Gel system

The extraction of the wheat grain endosperm storage proteins was conducted using sodium dodecyl sulfate, urea and 2-mercaptoethanol according to Lafiandra & Kasarda (1985). Crushed wheat grains were added to a 1.5 ml tube with 0.4 ml of the extraction solution, and the mixture was incubated overnight at room temperature. Extracts were used directly for SDS-PAGE electrophoresis according to Laemmli (1970) with some modifications (Payne, Holt & Law, 1981) to separate the wheat storage proteins.

Gel preparation and electrophoresis

The following stock solutions were prepared for the separating gel: (A) acrylamide, 40 g: Bis, 0.5 g; and ddH2O to 100 ml. (B) Tris. 6.15 g; SDS, 0.1411 g; dissolved with ddH2O, adjusted with HCL to pH 9.1 and brought to 100 ml. (C) Ammonium persulphate, 0.12 g dissolved in 5 ml ddH2O. Solutions A and B can be stored at 4 °C for a few weeks. Solution C was freshly prepared for each use.

For casting two separating gels, the following solution was prepared to produce a 10% (w/v) gel with 0.125% (w/v) crosslinking: 18 ml of solution A, 51 ml of solution B, 3 ml of solution C and 40 µl TEMED. The 5% separating gel with 0.26% crosslinking was as for the 10% gel except that the acrylamide stock solution A contained 20 g acrylamide and 1.04 g Bis. The solution was poured between the glass plates (25 × 11 cm with 1.5 mm spacers) to within two cm of the top. After polymerization, the unpolymerized material at the top of the gel was carefully removed with a syringe.

The 3% stacking gel was prepared by mixing 7.5 ml of acrylamide stock solution (Acrylamide, 6 g; Bis, 0.5 g; and ddH2O added to 100 ml), 6 ml of Tris-phosphate-SDS stock solution (Tris.1, 817 g; SDS, 0.25 g; dissolved ddH2O; adjusted with Ha PO to pH 6.7 and brought to 100 ml), 0.87 m1 ddH2O, 0.63 ml of the 2.4% ammonium persulfate and 20 µl TEMED. The stacking gel was poured onto the separating gel and the combs were quickly inserted because polymerization is rapid. After 15–20 min, the combs were removed gently to avoid damaging the slots. Slots must be free of any remaining polyacrylamide solution or polymerized debris.

10 µl of the sample extract was loaded into each gel slot with a pipette. Electrophoresis was performed in a DESAPHOR VA 150 vertical slab gel apparatus containing two gels. The high molecular weight glutenin subunits and gliadin bands were identified according to the nomenclature of Payne & Lawrence (1983) and Bushuk & Zillman (1978) by reference to Chinese Spring standard wheat.

Staining, destaining and drying of gels

Following electrophoretic separation, gels were stained overnight using the staining solution (0.01% (w/v) Coomassie blue R and 0.003% (w/v) Coomassie blue G dissolved in 18% (v/v) methanol). Gels were then destained in dH2Ofor 2–4 days. The water was changed two times daily until the desired contrast to the background was reached. After being scored, the gels were dried for preservation. The electrophoretic variants of gliadins and glutenins were labeled alphabetically according to the mobilities of their protein bands in SDS-PAGE.

Multiple sequence alignment

The sequences amino acids of durum wheat and bread wheat (File S1) were retrieved from the NCBI GenBank database. The sequences of ω, γ, and α gliadins were subjected to multiple amino acid sequence alignment using the CLUSTAL O (1.2.4) (https://www.ebi.ac.uk/Tools/msa/clustalo/) after manual adjustment, to determine the similarity regions and evolutionary relationships between the sequences.

Computational analysis

The chemical parameters of ω, γ, and α gliadins werein silico computed using the ProtParam tool (https://web.expasy.org/protparam/) for clarifying properties include the amino acid composition, molecular weight, aliphatic index, theoretical isoelectric point (pI), negatively and positively charged residues, and instability index (Walker, 2005).

Data analysis

The Protein bands were scored as absent (0) or present (1), each of which was treated as independent. The percentage of polymorphism (%) was estimated by dividing the polymorphic bands by the total number of scored bands. The clustering analysis was performed using R statistical software version 4.1.1, library factoextra (Kassambara, 2016) and the R codes have been uploaded as File S2.

Results

Genetic polymorphism and clustering analyses

The endosperm protein banding patterns (gliadins and glutenins) obtained by SDS-PAGE of the durum and bread wheat genotypes are presented in Fig. 1 (Figs. S1 and S2). The identification of the proteins on the gel is putative. Additional analytical techniques, such as peptide mass fingerprinting (PMF), would be necessary to confirm the identification of the proteins.

The analysis of SDS-PAGE patterns of durum wheat endosperm storage protein produced 16 loci, nine of which were polymorphic, while seven were monomorphic File S3. The polymorphism revealed by the SDS-PAGE was 56%. The phylogenetic relationship among the 23 durum genotypes was determined based on the SDS-PAGE banding profiles. Phylogenetic analysis divided the 23 durum wheat genotypes into five clusters. Durum wheat 22 genotype formed the cluster I independently. The cluster analysis grouped durum genotypes 5 and 6 in cluster II. Cluster III included genotypes 1, 2, 7, 8, 10 and 12. Genotypes 3, 15, 16, 17, 18, 19, 20 and 21 were grouped in cluster IV. Cluster V consisted of six genotypes; 4, 9, 11, 13, 14 and 23 (Fig. 2A). The SDS-PAGE patterns of bread wheat endosperm storage protein produced 18 loci, nine of which were polymorphic, while nine were monomorphic. The polymorphism revealed by the SDS-PAGE was 50%. The phylogenetic analysis (Fig. 2B) determined based on the SDS-PAGE banding profiles divided the 23 bread wheat genotypes into five clusters. Bread wheat genotypes; 13, 14, 15 and 17 formed the cluster I. The cluster analysis grouped bread genotypes 1, 9 and 18 in cluster II. Cluster III included genotype 23. Genotypes 10, 20, 21 and 22 were grouped in cluster IV. Cluster V consisted of genotypes; 2 and 3. Cluster VI included nine genotypes; 4, 5, 6, 7, 8, 17, 12, 16 and 19 (Fig. 2B). Phylogenetic analysis (Fig. 2C) based on the combined ω, γ, and α-gliadins profiles of both durum and bread wheat genotypes divided the 46 genotypes into seven clusters. Interestingly, the fifth and seventh groups included both durum and bread wheat genotypes.

Figure 1 Different glutenine and gliadin banding patterns in the regions of ω, γ, and α gliadins were observed in the genotypes studied.

(A) Glutenine and gliadin banding patterns of the 23 durum whaet genotypes. (B) Glutenine and gliadin banding patterns of the 23 bread wheat genotypes.

Figure 2 Clustering dendrogram based on gliadin and glutenine banding patterns.

(A) Phylogenetic relationship among 23 durum wheat genotypes. (B) Phylogenetic relationship among 23 bread wheat genotypes. (C) Combined phylogenetic relationship among 46 durum and bread wheat genotypes.

In silico analysis of ω, γ, and α-gliadins

In the current study, the amino acid sequences of the ω, γ, and α-gliadins were aligned to determine the intra-diversity at the protein level and to provide a possible reconstruction of phylogenetic relationships among durum and bread wheat. Sequence alignment showed that ω-gliadins of durum wheat share 47.9% identity and 52.1% polymorphism with ω-gliadins of bread wheat (Fig. 3). The alignment showed that the γ-gliadin protein sequence of durum wheat showed an identity of 93.4% and 6.6% polymorphism with γ-gliadins of bread wheat (Fig. 3). The alignment also showed that the α-gliadin protein sequence of durum wheat showed an identity of 84.6% and 15.4% polymorphism with α-gliadins of bread wheat (Fig. 3).

Figure 3 Alignments of the sequences of ω , γ , and α-gliadins of durum and bread wheat.

Computational chemical analysis

ω-gliadin

The comparison of amino acid residues (File S4) of ω-gliadin between durum wheat and bread wheat (Fig. 4A) indicated that bread wheat contained a higher content of glutamine (47.9%) and phenylalanine (10.1%), and a lower content of proline (18.6%) than the content of glutamine (37.6%) and phenylalanine (8.6%), and proline (26.7%) of durum wheat. The computational analysis indicated that the negatively (Asp + Glu) and positively (Arg + Lys) charged residues in the ω-gliadin of bread wheat were higher than those of durum wheat (Table 2). The analysis also revealed that the ω-gliadin of bread wheat possesses a higher aliphatic index (135.18), and lower theoretical pI (6.39), and an instability index (34.21) than the ω-gliadin of durum wheat (36.16), (8.12) and (156.99), respectively.

Figure 4 Amino acid composition of durum and bread wheat gliadins.

(A) ω-gliadins. (B) γ-gliadins. (C) α-gliadins.

Table 2 The computed protein parameters; protein size, molecular weight, extinction coefficient and instability index of ω, γ, and α-gliadins of durum and bread wheat.

	Triticum turgidum	Triticum aestivum	
	Omega gliadin	Gamma gliadin	Alpha gliadin	Omega gliadin	Gamma gliadin	Alpha gliadin	
Protein size (aa)	359	275	294	328	274	292	
Protein MW (KD)	41.770	31.470	33.731	39.631	31.450	33.612	
(Asp + Glu)	6	5	5	9	5	5	
(Arg + Lys)	7	9	6	8	9	7	
Aliphatic index	36.16	71.89	67.31	135.18	102.26	121.68	
Instability index	156.99	104.44	114.32	34.21	69.31	70.79	
Theoretical pI	8.12	8.70	7.62	6.39	8.72	8.30	

γ-gliadin

The comparison of amino acid residues (File S4) of γ-gliadin of durum wheat and bread wheat (Fig. 4B) revealed that bread wheat contained a higher content of glutamine (32.8%) and a lower content of proline (15%) than the content of glutamine (31.6%) and proline (16%) of durum wheat. The analysis of protein parameters indicated that the negatively and positively charged residues in the γ-gliadin of bread wheat were equal to those of durum wheat (Table 2). The computational analysis also showed that the γ-gliadin of bread wheat possesses a higher aliphatic index (102.26), theoretical pI (8.72), and lower instability index (69.31) than the γ-gliadin of durum wheat (71.89), (8.7) and (104.44), respectively.

α-gliadin

The comparison of amino acid residues (File S4) of α-gliadin of durum wheat and bread wheat (Fig. 4C) indicated that bread wheat contained a lower content of glutamine (33.2%) and a higher content of proline (15.4%) than the content of glutamine (36.1%) and proline (14.3%) of durum wheat. The c analysis indicated that the negatively and positively charged residues in the α-gliadin of bread wheat were higher than those of durum wheat (Table 2). The computational analysis also showed that the α-gliadin of bread wheat possesses a higher aliphatic index (121.68), theoretical pI (8.30), and lower instability index (70.79) than the α-gliadin of durum wheat (67.31), (7.62) and (114.32), respectively.

Discussion

Diversity in the composition of the putative gliadin fraction has proved useful for cultivar identification (Mefleh et al., 2020; Lavoignat et al., 2022). An additional criterion used for assessing the intra-genetic diversity is the in silico and computational analyses for the sequences of seed proteins such as ω, γ, and α-gliadins. Differences in electrophoretic banding patterns between and within the durum and bread wheat genotypes were observed with the main variation being in the ω, γ, and α-gliadins polypeptides. In the current study, the percentage of polymorphism resulted from the SDS-PAGE of among the 23 genotypes of the durum wheat ω, γ, and α-gliadins (56%) were higher than those revealed by the SDS-PAGE among the 23 genotypes of bread wheat (50%). The SDS-PAGE is utilized to assess the polymorphism in various plant cultivars (Suvorova & Kornienko, 2011; Rayan & Osman, 2019) and to evaluate the genetic variation of inter- and intra-specific wheat genotypes (Abou-Deif, Khattab & Afiah, 2005; Lata et al., 2017; Sen, Biswas & Sinha, 2021). The clustering analysis based on the SDS-PAGE banding profiles divided the 23 durum genotypes into five clusters. In the same context, the 23 bread genotypes were divided into six clusters. Also, the analysis based on the combined ω, γ, and α-gliadins profiles of both durum and bread wheat genotypes divided the 46 genotypes into seven clusters due to the percentage of similarity presented in the SDS-PAGE profiles. The results presented here clearly showed that SDS-PAGE can be used effectively and simply to assess the genetic variation of wheat cultivars. The differences found between and within the two species in profiles of putative gliadins may be due to the genetic background currently utilized in breeding programs.

In addition, differences in density for apparently equivalent protein components were occasionally noted in the patterns of different genotypes. Such intensity differences may be due to differences in the frequency of gene transcription as a consequence of differences in noncoding DNA that has a controlling function, e.g., promoter sequences. Differences in the number of active gene copies among varieties could change the amount of protein synthesized during endosperm development, thus affecting the intensity of the equivalent band in the electrophoretic pattern (Cavalier-Smith, 1978; Al-Khayri et al., 2023b).

Multiple alignments of ω, γ, and α-gliadins sequences of durum wheat share 52.1%, 6.6% and 15.4% polymorphism with ω, γ, and α-gliadins sequences of bread wheat, respectively. The results indicated that the vast majority of differences between durum and bread wheat based on gliadins profiles were due to the ω-gliadin and the presence of the repetitive regions and the gaps in the interior of the repetitive domains that are mainly responsible for the size of heterogeneity of the ω-gliadins. Some studies reported that SNPs and changes in the repetitive region are responsible for the sequence diversity between the -gliadin genes (Anderson, Hsia & Torres, 2001).

According to the in silico analysis of ω, γ, and α-gliadins parameters, the bread wheat possessed the highest content of glutamine, negatively and positively charged residues, and aliphatic index. This result may explain the importance of glutamine, aspartic acid, glutamic acid, arginine, lysine residues, and aliphatic index in bread-making characteristics of bread wheat. In the same context, the durum wheat possessed a higher instability index and proline residue. This finding may explain why durum wheat is preferred in the making of pasta due to its high elasticity-related content. The result revealed that ω and γ-gliadins possess higher content of glutamine residue and lower content of proline residue, while α-gliadin possesses higher proline and lower glutamine content in bread wheat than durum wheat.

It is well recognized that the cysteine residues contribute significantly to the distinctive qualities of wheat flour, and as a result, they are crucial for the quality of the dough. According to the primary structure of ω, γ, and α-gliadins, the ω-gliadin is free of cysteine residues in both durum and bread wheat. The typical γ-gliadin contains eight cysteine residues. In this study, it was found that γ-gliadins contain eight cysteine residues in both durum and bread wheat. Furthermore, we identified six cysteine residues in α-gliadins of durum wheat and five cysteine residues in α-gliadins of bread wheat. This may make the dough of durum wheat much stronger than the dough of bread wheat (Ikeda et al., 2002; Masci et al., 2002). Variations in the numbers and position of cysteine residues may have an impact on the disulfide bond formation pattern, failing to create some intramolecular disulfide bond(s). The production of intermolecular disulfide linkages and the construction of polymers would then be possible with these cysteine residues (Masci et al., 2002).

Conclusion

Assessing the genetic diversity and the relationship inter- and intra-different wheat genotypes using the SDS-PAGE technique has become a simple, effective and cheap approach to estimating the genetic diversity of the wheat genotypes which helps breeders develop their crossing program instead of many expensive DNA techniques. The SDS-PAGE profiles of putative durum wheat gliadins revealed 56% and 50% polymorphism between the durum wheat and bread wheat genotypes, respectively. The intra-genetic diversity of ω, γ, and α-gliadins of durum wheat revealed 52.1%, 6.6% and 15.4% with ω, γ, and α-gliadins of bread wheat, respectively. In both ω, and γ-gliadins, the bead wheat possessed higher content of glutamine, aspartic acid, glutamic acid, arginine, lysine residues and aliphatic index that may explain their importance in bread-making characteristics in bread wheat. The durum wheat possessed a higher instability index and proline residue that may explain the preference for durum wheat in pasta production. The α-gliadin possessed higher proline and lower glutamine content in bread wheat than in durum wheat. So, the selection of high quality bread wheat genotypes should be based on the presence of high ω and γ–gliadins content while the selection of durum wheat for pasta production should be based on the presence of α-gliadin content.

Supplemental Information

Supplemental Information 1 Glutenine and gliadin banding patterns of 23 durum genotypes

Click here for additional data file.

Supplemental Information 2 Glutenine and gliadin banding patterns of 23 aestivum genotypes

Click here for additional data file.

Supplemental Information 3 Gliadin sequences

Click here for additional data file.

Supplemental Information 4 Intra-genetic diversity 0,1

Click here for additional data file.

Supplemental Information 5 Amino acid composition

Click here for additional data file.

Supplemental Information 6 R codes

Click here for additional data file.

Additional Information and Declarations

Competing Interests

Author Contributions

Data Availability

Diaa Abd El-Moneim is an Academic Editor for PeerJ.

Jameel M. Al-Khayri conceived and designed the experiments, prepared figures and/or tables, and approved the final draft.

Khairiah M. Alwutayd conceived and designed the experiments, prepared figures and/or tables, and approved the final draft.

Fatmah A. Safhi conceived and designed the experiments, authored or reviewed drafts of the article, and approved the final draft.

Mesfer M. Alqahtani performed the experiments, prepared figures and/or tables, and approved the final draft.

Rana M. Alshegaihi performed the experiments, authored or reviewed drafts of the article, and approved the final draft.

Diaa Abd El-Moneim conceived and designed the experiments, prepared figures and/or tables, and approved the final draft.

Shri Mohan Jain performed the experiments, analyzed the data, authored or reviewed drafts of the article, and approved the final draft.

Ahmed S. Eldomiaty performed the experiments, analyzed the data, authored or reviewed drafts of the article, and approved the final draft.

Rahma Alshamrani conceived and designed the experiments, prepared figures and/or tables, and approved the final draft.

Amani Omar Abuzaid performed the experiments, prepared figures and/or tables, and approved the final draft.

Abdallah A. Hassanin analyzed the data, authored or reviewed drafts of the article, and approved the final draft.

The following information was supplied regarding data availability:

The raw data are available in the Supplemental Files.

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
