# Peer review of "Assessment of intra- and inter-genetic diversity in tetraploid and hexaploid wheat genotypes based on omega, gamma and alpha-gliadin profiles"

_PeerJ, doi:10.7717/peerj.16330_

## Round 0.1 · original submission · Major Revisions

Dear authors
According to the reviewer's comments, you should revise your manuscript and submit it again.
1. You should reorganize the introduction.
2. You need to emphasize your aim well.
3. You should include information about the genotypes used in the experiment.
4. It would be good to share the R codes.

Reviewer 1 ·

Basic reporting

Clear, technically correct but not entirely professional English is used throughout. For this reason, there is a need for a more professional editing of the writing language of the article.
The article lacks sufficient introduction and background to show how the work fits into the wider field of knowledge. Appropriate reference should be made to the relevant previous literature. It would be more useful to explain in more detail why it differs from the previous applications and why this method should be used.
The structure of the article can be said to conform to a generally acceptable form of "standard sections" (according to your Instructions for Authors).

Experimental design

This research is original primary research within the topics specified in the Purpose and Scope of the journal.
The research question is not well defined. Exactly how the research fills a defined knowledge gap is not specified. There are some uncertain situations in this respect. In this study, it was aimed to investigate the variation within and between durum and bread wheat genotypes. However, not enough information has been given about the nature of the genetic material to be used and experimental area and ecology.
The research must have been carried out meticulously and to a high technical standard and in accordance with the ethical standards applicable in the field.
The methods are described with sufficient information and detail so that they can be repeated by other researchers.
The research was carried out according to the randomized blocks experimental design in the growing season of 2021-2022. There is no need for repeated trials for such a study. It will only be sufficient to carry out the same agronomic practices in the same ecological conditions. This situation raises the suspicion that the study is more comprehensive and only a part of it is given in this article.

Validity of the findings

This study is a repetition of a known method. Such studies can also be repeated periodically. However, the rationale for repetition and how it adds value to the literature should be clearly explained and comparisons should be made. Otherwise, it would be more realistic to consider it as unnecessary or a derivative of the existing work.
It is understood that the data on which the results are based are obtained and used in a discipline-specific manner. The data are robust, statistically sound and controlled.
Can the result obtained as a result of the research be used only in determining the diversity? For example, have bread or durum wheat genotype(s) with very extra characteristics that can be used as parent (germplasm) in combination breeding have been determined?
In addition, suggestions and wishes should have been included in order to disseminate the obtained results.

Additional comments

Apart from my explanations above, as a result of my personal evaluation, I have come to the conclusion that the article can be published after major revision.

Annotated reviews are not available for download in order to protect the identity of reviewers who chose to remain anonymous.

Reviewer 2 ·

Basic reporting

no comment

Experimental design

no comment

Validity of the findings

no comment

Additional comments

There are some ambiguous points in the scope of the aim. Although it is possible to separate the individuals of genotypes through other molecular techniques such as DNA-marker assistant selection technique, why the authors used this techniques is unclear. The genotype information for hexaploid or tetroploid wheat is not exist and there is. Moreover, the article reevaluates the current literature knowledge.

I suggest R codes should be shared for clarity of the data represent the dendrogram generated by clustering analysis. The captions of the supplementary table need to be added. Statistical significance degrees and models used in the analysis should be given on the main text.

·

Basic reporting

This is a revision of the manuscript titled “Assessment of intra- and inter-genetic diversity in
tetraploid and hexaploid wheat genotypes based on omega, gamma and alpha-gliadin profiles (#83389)”. The manuscript was related to the omega, gamma and alpha-gliadin variations in durum and bread wheat.

The English is overall ok, I have listed below some words/sentences that needs to be corrected (but there are probably others that I did not notice in my revision). It needs English editing by a fluent English speaker.

Introduction is poorly written. There are only two paragraphs in the introduction and sentence/paragraphs are not logically connected.

In the first paragraph, many topics such as the importance of wheat, its use, genome structure, genetic diversity, information about gliadins and previous research are given together. In the first paragraph, the importance of the subject and the research question should have been given and/or should have been explained why the research was conducted.

Other topics such as (1) genome structure and genetic diversity of durum and bread wheat, (2) information about gliadins and previous studies should be written in the next paragraphs.

Some sentences below are unrealistic.

Lines 55-58 The combined … the genetic diversity.
The vast majority of genetic diversity existing today in wheat germplasms is caused by mutation, recombination, immigration of genes and interspecies spontaneous hybridization, not modern techniques.

Lines 73-74 However, there … species of wheat.
There are hundreds of studies that analyze genetic diversity in durum and bread wheat using the SDS page.

Experimental design

In material and method, authors used 23 tetraploid and 23 hexaploid wheat genotypes from CIMMYT. The use of genotypes from ICARDA will increase the international importance of study. But no information was given about the plant material and it is not explained why these genotypes were selected.

The SDS PAGE preparation procedure was unnecessarily detailed. It could be simply cited, and if any modifications could be specified. There is no protein molecular weight marker in gels. It is not specified how the high and low molecular weight bands were determined. Molecular weight of different band was not detected.
It was not specified how the amino acid sequences are determined and which genotypes are used the determination of amino acid sequences.

Validity of the findings

Results
Your writing is wordy. You can trim the text by good 25% by being more efficient avoiding repetitions. The first sentence (line 176-178) of results is a pure repetition from material and method; please remove it.

For example, remove the sentences in the lines 211-211, 222, 232. You can write at first as “Alpha, gamma and -gliadin were rich in glutamine and proline”

Computational analysis results of omega, gamma and alpha gliadins (210-229) are written almost exactly the same. This is very routine, if the results must be given similarly, you can give it under one title.

Discussion

This section should be well revised and the results need to be discussed better
Lines 241-250. the sentences should go to the introduction. Because it does not contain information that supports any results identified in this study.
line 241. The “plant breeding” can maximize
Line 242. Remove “t”
Lines 252-255. This sentence is a pure repetition from results. It must be removed.

Lines 267-270. Protein band profiles do not differ by environment. In the research already mentioned (Abou-Deif et al. 2005), changing by environment is the protein ratio, not the protein band profile.
Besides, it's not a good way to use in discussion this sentence “the results in agreement with other studies”. Simply discuss the reasons for the results of your research.
Lines 271-272. What is “intensity”. If you mean the density band (for example, α/β Gliadin bands of Durum9 and 10 genotypes in Figure 1a) in gels, please specify.
Lines 286-291. This sentence (line 176-178) is a repetition from results and did not include any discussion.
Line: 289 Please remove “important” and “from this analysis is”
Multiple alignment results for cysteine residue are well discussed (Lines 292-303). However, there is no discussion for other amino acid residues.
The following sentences should be supported by citation.
Lines 274 -276.
Lines 282-293.

Conclusion
This section should include your inference and conclusions from the research results, and you should give recommendations for researchers on this topic and suggestions for future research.

Lines 308-315 This is a pure repetition of results.

Figures

Figures are relevant, well labelled and described.

Additional comments

I think that the manuscript may not currently be published in a Q2 ranked journal. Introduction must be reorganized with additional two paragraph. The originality of the research is the comparison of amino acid sequences of gliadins in durum and bread wheat. Methodology about determination of amino acid sequences of gliadins is not given in the material method. It is not clear whether author determined the sequences in this research or used the existing sequences. Discussion and conclusion sections should be revised and the results need to be discussed better. I recommend the authors to revise the manuscript carefully and resubmit again.

---

## Round 0.2 · Minor Revisions

Dear authors
After the corrections you made, I sent your manuscript back to one of the reviewers for review and he/she informed me that some minor changes were needed. Once you make the changes and complete the deficiencies, your manuscript is acceptable.

·

Basic reporting

no comment

Experimental design

no comment

Validity of the findings

no comment

Additional comments

The authors have made significant improvement in the article, but I have suggestions. The manuscript can be published after minor revision.

---

## Round 0.3 · Minor Revisions

Dear Authors,

I appreciate your effort in improving your manuscript by making changes. However, you need to add the pedigrees of the material you used in the material method section as a table. This is my request and the last reviewer's. Please add this information, if you cannot add it, indicate the reason. The pedigree information of advanced inbred lines is important information. It must be included in the manuscript.

Best regards

---

## Round 0.4 · Minor Revisions

Dear authors,

Please edit your manuscript to make it clear that the identification of the proteins on the gel is "putative" and acknowledge that to really prove their ID, additional analyses such as a peptide mass fingerprint would be necessary.

I do not expect you to perform the additional analysis (though you are welcome to do so if you have the facilities) - simply to state this as a limitation of the current work.

Best regards

---

## Round 0.5 · Minor Revisions

While reviewing the final revised version of your manuscript, I noticed another shortcoming that I had overlooked before.

One of the reviewers asked this, but it must have escaped my attention. You did not specify the marker (ladder) you use in your gels. When marking regions of gliadin or glutenin on the gel, you should have calculated them according to a certain molecular weight ladder marker. You could also use relative mobility instead. The most well-known marker is Neepawa. But there are others (Chinese Spring, Marquis ect.). Thus, you could separate the gliadin regions according to their molecular weights by comparing them with the ladder marker or by relative mobility values. Please add this.

If you haven't used a ladder marker, clearly state that as well. In this case, you will need to explain how you detected the gliadin and glutenin regions. If you have used ladder markers, you may not need to specify that the regions are the default. With appropriate literature support, you can say that we have separated them according to molecular weight or relative mobility values.

Sorry I realized this late, but this is an important shortcoming. Please correct and explain.

---

## Round 0.6 · Minor Revisions

Dear Authors

You added to the manuscript that you based it on the catalogs of Payne (1983) and Bushuk and Zillmann (1978). In your response letter, you stated that Chinese Spring could be used as a standard wheat. However, these catalogs were not created only for Chinese Spring. There are other wheat varieties out there and they are used as standard wheat varieties for the presence of different glutenin regions. There are 1, 2, 2star, 7+8, 2+12 etc. bands in HMW-GS, each located a different allele. Of course, you cannot show them this way and you also stated that they are "putative". However, I am wondering how you compare the presence and absence of bands. Which band did you use as reference? Because normally the Rm value of the thickest band in standard wheat is considered to be approximately 50, and by measuring its distance to the beginning of the gel, the Rm values of other bands in the genotypes are calculated and their differences are determined so that we can say whether they are present or absence. How did you do it? Therefore, I assumed that you were referring to the band pattern of a wheat variety from this catalog you mentioned and that it was Chinese Spring, as you stated in your reply letter. I think it might be appropriate to add the following sentence to your manuscript.
"The high molecular weight glutenin subunits and gliadin bands were identified according to the nomenclature of Payne and Lawrence (1983) and Bushuk and Zillman (1978) by reference to Chinese Spring standard wheat."

If you disagree with my comment, feel free to state so clearly.

Best regards

---

## Round 0.7 · accepted · Accept

Thank you for your revision. Your manuscript is now acceptable.